# Targeting the latent cytomegalovirus reservoir with an antiviral fusion toxin protein

B.A. Krishna[1], K. Spiess[2], E.L. Poole[1], B. Lau[1,†], S. Voigt[3,4], T.N. Kledal[5], M.M. Rosenkilde[2] & J.H. Sinclair[1]

Reactivation of human cytomegalovirus (HCMV) in transplant recipients can cause life-threatening disease. Consequently, for transplant recipients, killing latently infected cells could have far-reaching clinical benefits. *In vivo*, myeloid cells and their progenitors are an important site of HCMV latency, and one viral gene expressed by latently infected myeloid cells is *US28*. This viral gene encodes a cell surface G protein-coupled receptor (GPCR) that binds chemokines, triggering its endocytosis. We show that the expression of US28 on the surface of latently infected cells allows monocytes and their progenitor CD34 + cells to be targeted and killed by F49A-FTP, a highly specific fusion toxin protein that binds this viral GPCR. As expected, this specific targeting of latently infected cells by F49A-FTP also robustly reduces virus reactivation *in vitro*. Consequently, such specific fusion toxin proteins could form the basis of a therapeutic strategy for eliminating latently infected cells before haematopoietic stem cell transplantation.

[1] Department of Medicine, Addenbrooke's Hospital, University of Cambridge, Cambridge CB20QQ, UK. [2] Laboratory for Molecular Pharmacology, Department of Neuroscience and Pharmacology, Faculty of Health and Medical Sciences, University of Copenhagen, Copenhagen DK-2200, Denmark. [3] Department of Infectious Diseases, Robert Koch Institute, Nordufer 20, Berlin 13353, Germany. [4] Department of Pediatric Oncology/Hematology/SCT, Charité-Universitätsmedizin, Berlin 13353, Germany. [5] Section for Virology, The National Veterinary Institute, Technical University of Denmark, Frederiksberg DK-1870, Denmark. † Present address: Medical Research Council-University of Glasgow Centre for Virus Research, Glasgow G61 1QH, United Kingdom. Correspondence and requests for materials should be addressed to B.A.K. (email: back2@cam.ac.uk) or to J.H.S. (email: js@mole.cam.ac.uk).

Human cytomegalovirus (HCMV) is a species-specific β-herpesvirus that has characteristic lytic and latent stages to its life cycle[1]. It is a ubiquitous pathogen, infecting 50–90% of populations, depending on socioeconomic factors, maintaining a lifelong infection in the carrier[2]. Primary HCMV infection is normally well controlled by a robust humoral and T cell-based immune response and, consequently, it is often asymptomatic in healthy individuals. Despite this, primary HCMV infection is never cleared by the immune system and results in a lifelong viral persistence that is underpinned by the ability of the virus to undergo latent infection. Although sporadic reactivation from latency likely routinely occurs *in vivo*[3], it is asymptomatic in healthy individuals. However, virus reactivation, like primary infection, poses a severe clinical threat to immunocompromised individuals, such as transplant recipients and patients with AIDS[4].

Although HCMV has a wide cell tropism for the lytic life cycle, its latent life cycle appears more restricted. One site of latent carriage is in cells of the early myeloid lineage that includes CD34+ progenitor cells and CD14+ monocytes[4]. In these latently infected cells, maintenance of viral genome is associated with expression of a relatively small number of latency-associated viral genes and no infectious virus is produced. This in itself likely aids these sites of latency to escape host immune surveillance[5]. It is now also clear that, in healthy HCMV-positive donor cells, differentiation of CD14+ monocytes and CD34+ progenitors to dendritic cells and macrophages triggers virus reactivation from latency[3,6–8] and that this differentiation-dependent reactivation of latent virus in the myeloid lineage is mediated by changes in post-translational modification of histones around the viral MIEP (major immediate-early promoter). This drives viral major immediate-early (IE) gene expression, resulting in reactivation of the viral lytic gene cascade and the production of infectious virions[7]. Latent infection is, therefore, a key aspect of viral persistence and latently infected cells are a clear threat to the immunocompromised. Despite this, there have only been few reports describing the therapeutic targeting of latently infected cells[9].

For this reason, much attention has been drawn to viral gene products expressed during HCMV latency and the possibility that they might alter the phenotype of the latently infected cell that can then be exploited for novel antiviral therapies[5]. For instance, latent infection has been shown to induce changes in cellular RNA expression[10–13], the cellular secretome[14] as well as cell surface protein expression[9]. These changes are likely mediated by expression of the HCMV latency-associated genes[15,16]. A number of these have been detected in naturally latently infected cells from healthy seropositive donors, including *UL84*, *UL81-82ast*, *UL138*, *UL144*, *LUNA*, *UL111A*, *RNA4.9* and *US28* (refs 16–20). Although the functions of many of these latency-associated gene products are becoming better established, their roles in latency specifically are still unclear.

Allogeneic haematopoietic stem cell transplants (allo-HSCTs), for example, from granulocyte colony-stimulating factor (CSF)-mobilized donors, are routinely used as a treatment for several high-risk leukaemias and other nonmalignant diseases. Peripheral blood stem cells are now one of the most common sources of stem cells for allo-HSCT and comprise multipotent CD34+ cells that expand and differentiate to reconstitute the immune system. However, this differentiation can also result in HCMV reactivation in up to 80% of allo-HSCT patients, if not treated with antivirals[21]. Although prophylactic treatment with antivirals such as ganciclovir and foscarnet keeps CMV disease incidence below 10% in these patients, ganciclovir mediated neutropenia can lead to increased mortality from bacterial and fungal infections[22]. Consequently, the reduction in latent HCMV load in HSCTs could have far-reaching clinical benefits[23–27].

US28 is one of four G protein-coupled receptor (GPCR) homologues encoded by HCMV[28]. All four receptors are expressed during lytic infection[29,30], but only *US28* mRNA has been detected in models of latent infection[31–33] as well as naturally latently infected monocytes[34]. Similarly, recent work from Humby and O'Connor[35] has shown that *US28* is important to establish latency in CD34+ cells. US28 is also the best characterized of these virus-encoded receptors; it binds both CC and CX3C chemokines[36] and this ligand binding affects US28 constitutive signalling[37,38]. This appears to promote proliferative signals during lytic HCMV infection that, therefore, have been linked to vascular diseases and potential oncomodulatory effects[39–41]. US28 has also been shown to heteromerize with the other HCMV-encoded GPCRs UL33 and UL78 that inhibits constitutive US28 activation of nuclear factor-κB[42].

Fusion toxin proteins (FTPs) exploit high-affinity receptor–ligand interactions to direct cytotoxic molecules to target cells, and have shown success as novel cancer therapies[43,44]. Moreover, the technique has a great potential as a treatment for other indications, such as infectious diseases, where pathogen-encoded targets provide superior specificity[45]. Recently, a novel fusion toxin protein (F49A-FTP) has been described that targets and kills cells lytically infected with HCMV[46]. F49A-FTP is based on the soluble extracellular domain of the US28 ligand CX3CL1 (also known as fractalkine) and binds US28 with high affinity compared with the cellular CX3CL1 receptor, CX3CR1. After binding US28, F49A-FTP is internalized and mediates cell killing via a recombinant *Pseudomonas* exotoxin-A motif.

Here, we demonstrate that F49A-FTP is able to kill monocytes and CD34+ progenitor cells that are experimentally latently infected with HCMV and that this killing is dependent on US28 expression. Furthermore, we show that this killing is effective at reducing viral load in naturally latently infected CD14+ monocytes. Consistent with this reduction in latent load, this FTP robustly reduces the frequency of virus reactivation from experimentally and naturally latently infected cells. These results are, therefore, proof of principle that F49A-FTP can purge the latent load of HCMV in haematopoietic stem cell grafts that could form the basis for a novel approach to greatly reduce the clinical threat of HCMV-positive grafts in the stem cell transplant setting.

## Results

**F49A-FTP kills myeloid cells that express US28 in isolation**. It was previously shown that F49A-FTP is able to kill fibroblast cells that were lytically infected with HCMV[46]. To start, we wanted to demonstrate that this cytotoxicity was due solely to US28 expression and not because of other factors associated with viral infection. Consequently, we infected human foreskin fibroblasts (HFFs) with two isolates of HCMV; the wild type, clinal isolate, Titan strain of HCMV or Titan with a deletion in the US28 gene (Titan-ΔUS28), both of which have a green fluorescent protein (GFP)-tagged UL32 gene. Cell cultures were then treated with F49A-FTP for 72 h before infected cells were visualized by fluorescence microscopy. F49A-FTP was able to kill HFFs infected with wild-type Titan HCMV but not the corresponding US28-deletion virus (Fig. 1).

In addition to lytic infection, we also wanted to assess the ability of F49A-FTP to kill cells in which HCMV would normally establish latent infection. To do this, we used THP-1 cells, a monocyte-like cell line and can be used as a model of HCMV latency[47,48]. THP-1 cells were transduced with lentiviral vectors to express haemagglutinin (HA)-tagged US28 (HA-US28) in

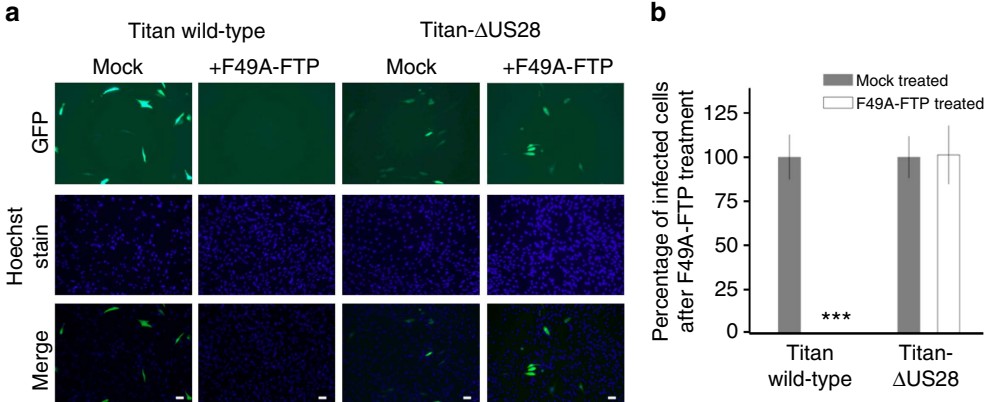

**Figure 1 | F49A-FTP kills lytically infected cells because of their expression of US28.** Human foreskin fibroblast cells (HFFs) were infected with either HCMV Titan wild-type or HCMV Titan-ΔUS28 at an MOI of 0.1. Both viral isolates have a UL32-GFP tag, causing infected cells to appear green by fluorescence microscopy. Cultures were then either mock-treated with PBS or treated with $5 \times 10^{-8}$ M F49A-FTP for 72 h and observed by fluorescence microscopy. (**a**) Representative images of the virally infected cultures with or without F49A-FTP. (**b**) A graphical representation of these data. Cell numbers were quantified by Hoechst staining cell nuclei, and the percentage of infected (green) cells is shown as a percentage of the control. White bars indicate 50 μm scale. Means and error bars (showing s.d.) were generated from three independent experiments. Statistical analyses were carried out using a paired two-tailed t-test and ***$P = 0.001$ were considered significant.

isolation. After this, puromycin selection was used to generate a cell population expressing HA-US28, and HA-US28 expression was confirmed by immunoblot analysis (Fig. 2a). Incubation of these THP-1 cells with F49A-FTP for 48 h showed that F49A-FTP efficiently killed US28-expressing THP-1 cells but not mock-transduced cells (Fig. 2b).

**Latently infected monocytes express cell surface US28.** On the basis that F49A-FTP is able to kill myeloid cells expressing US28 in isolation, we reasoned that this could enable us to target latently infected myeloid cells that are known to express US28 mRNA[34,49]. However, in order for this approach to work, it was important to confirm that US28 protein is expressed and trafficked to the cell surface during latent infection of CD14+ monocytes. Because of the lack of available anti-US28 antibodies capable of detecting US28 in latent cells, we were unable to stain latently infected cells directly. Consequently, we turned to radiolabelling assays. We latently infected CD14+ monocytes with an HCMV TB40E isolate with an SV40-GFP tag (SV40-GFP-TB40E)[50] that allows the short-term detection of GFP-tagged latently infected cells[9] and sorted GFP-positive monocytes to enrich for a population of latently infected monocytes by fluorescence-activated cell sorting. We confirmed by reverse transcriptase-quantitative PCR (RT-qPCR) analysis of HCMV gene expression that these cells were latent (Supplementary Fig. 1a). RT-qPCR of these cells before and after sorting showed high levels of expression of UL138 mRNA, a known latency-associated gene, concomitant with low levels of lytic transcripts (IE and UL99) compared with reactivated mature dendritic cells that resulted in a characteristic lytic transcription profile, with a switch to high levels of lytic IE and late RNAs (Supplementary Fig. 1a), consistent with previous studies[51–55]. We also analysed viral IE and late protein expression in these cells (Supplementary Fig. 1d) and observed no IE or UL99 protein expression. Similarly, no infectious virus could be isolated from these latently infected cells (Supplementary Fig. 1e), as expected. In contrast, reactivation of these latently infected monocytes resulted in detectable IE and UL99 expression (Supplementary Fig. 1d) as well as production of infectious virions (Supplementary Fig. 1e).

We then measured the affinity of sorted, latently infected cells for [125]iodine-radiolabelled CX3CL1, which is a high-affinity

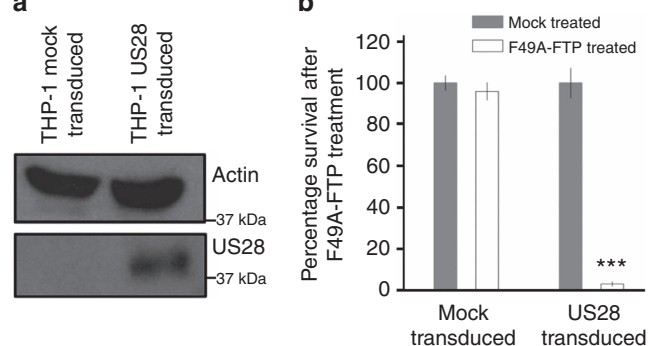

**Figure 2 | F49A-FTP kills monocyte-like THP-1 cells that express US28.** THP-1 cells expressing HA-US28 were generated by lentiviral transduction and puromycin selection. (**a**) Western blots of cell lysates from transduced THP-1 cells against the N-terminal HA tag on HA-US28, control cells were mock transduced. (**b**) THP-1 cells expressing HA-US28, or control cells, were treated with $5 \times 10^{-8}$ M F49A-FTP for 48 h. Cell death was quantified by Trypan blue staining. Means and error bars (showing s.d.) were generated from seven independent experiments. Statistical analyses were carried out using a paired two-tailed t-test and ***$P = 0.001$ were considered significant.

ligand of both US28 (ref. 36) and the endogenous receptor CX3CR1, that is expressed on monocytes. Figure 3a,b show that latently infected monocytes showed a higher binding affinity for CX3CL1 compared with uninfected monocytes, similar to the increase observed in US28 expressing Cos 7 cells compared with COS-7 cells expressing CX3CR1. We considered that although this result could result from latency-associated expression of US28, formally it could also have been because of increased expression of CX3CR1, perhaps induced by latent infection. To differentiate between these two possibilities, we performed RT-qPCR analysis over several time points post infection, measuring relative US28 and CX3CR1 expression (Fig. 3c,d). This showed that US28 is expressed from 1 day post infection in our model of latency. It also showed that, although latent viral infection does cause a minor increase in CX3CR1 expression, at 5 days post infection, when the binding assays were carried out in

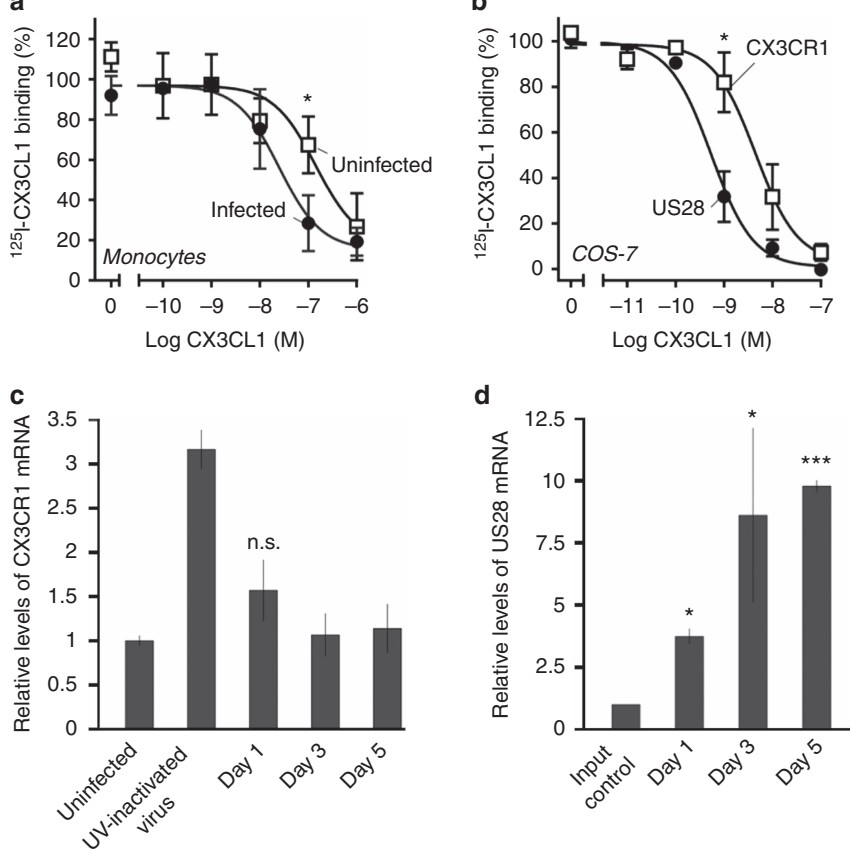

**Figure 3 | Latently infected monocytes have high affinity for CX3CL1 that indicates cell surface US28 expression.** Homologous competition binding on COS-7 cells and monocytes. (**a**) Binding of $^{125}$I-CX3CL1 to uninfected (white symbols) and sorted, GFP-positive (latent) HCMV-infected monocytes (black symbols). The data are normalized to maximal binding on infected cells. Error bars indicate s.e.m. for five independent biological replicates. (**b**) Binding of $^{125}$I-CX3CL1 to transiently transfected COS-7 cells expressing US28 (black) and CX3CR1 (white symbols). The data are normalized to maximal specific binding on US28-expressing cells. Error bars indicate s.e.m. for three independent biological replicates. (**c**) RT-qPCR analysis of CX3CR1 mRNA expression in monocytes latently infected at an MOI of 5, with SV40-GFP-TB40E, or monocytes treated with UV-inactivated virus, relative to an uninfected control. UV-inactivated virus samples were taken 1 day post infection. Means and s.d. values are shown from three measurements and normalized to GAPDH. (**d**) RT-qPCR analysis of US28 mRNA expression in monocytes latently infected at an MOI of 5, with SV40-GFP-B40E, relative to mRNA harvested immediately after infection (input control). Means and s.d. values are shown from three measurements and normalized to GAPDH. Statistical analyses were carried out using a paired two-tailed $t$-test and P values expressed as $*P = 0.05$ or $***P = 0.001$ were considered significant.

Fig. 3a, CX3CR1 showed no significant increase in expression. Taken together, these data are consistent with the view that US28 is expressed on the cell surface of latently infected monocytes.

**F49A-FTP kills experimentally latently infected monocytes.** Knowing that latently infected cells do, indeed, bind CX3CL1 with higher affinity than control monocytes, as a result of US28 expression, we next analysed whether F49A-FTP could kill latently infected cells. To do this, we experimentally infected CD14 + monocytes with SV40-GFP tagged HCMV[50], confirmed latency by RT-qPCR (see Supplementary Fig. 1b) and tested the effect of treatment with F49A-FTP. We based the concentration of F49A-FTP used ($5 \times 10^{-8}$ M) and the incubation time of 72 h on pilot titration experiments on latently infected monocytes (Supplementary Fig. 2).

After 72 h of treatment with F49A-FTP, we observed a significant loss of GFP-positive, latently infected monocytes (Fig. 4a). In order to ensure that these observations were because of the killing of latently infected cells, and not just a result of silencing of the SV40-GFP expression cassette, we next removed

F49A-FTP from these cultures and replaced it with fresh media containing cytokines followed by lipopolysaccharide (LPS) in order to stimulate differentiation and maturation to mature dendritic cells (mDCs), thereby resulting in reactivation of any remaining latent viral genomes. As the SV40-GFP cassette in the viral genome is also expressed during lytic infection[50], we again quantified the number of GFP-expressing mDCs resulting from reactivation of GFP-tagged virus. Figure 4b shows that the percentage of GFP-positive CD14 + monocyte-derived mDCs was also reduced in F49A-FTP-treated cultures. Finally, we co-cultured these CD14 + monocyte-derived mDCs with HFFs and quantified foci of viral IE expression by immunofluorescent staining in the fibroblasts as a measure of the reactivation of infectious virions (Fig. 4c) and again observed that F49A-FTP treatment resulted in a significant reduction in the amount of reactivated infectious virus. Taken together, these data show that F49A-FTP can target and reduce the number of experimentally latently infected CD14 + monocytes thus reducing subsequent reactivation of latent virus after differentiation and maturation.

If this targeting of latently infected cells was specifically because of US28 expression, F49A-FTP should be less effective against monocytes infected with a US28 deletion virus. To address this,

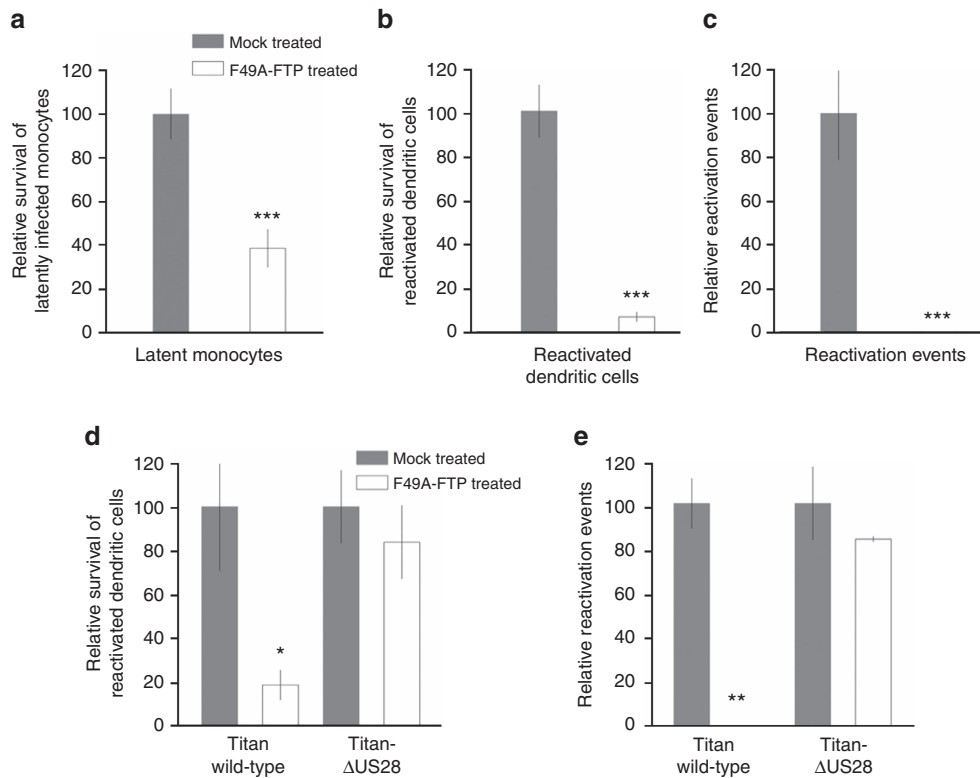

**Figure 4 | F49A-FTP kills experimentally latently infected monocytes, reducing HCMV reactivation events.** CD14+ peripheral blood monocytes were isolated and experimentally infected at an MOI of 5 with HCMV SV40-GFP-TB40E wild type that led to a mean average of 10.9% latently infected cells. After 24 h, F49A-FTP was added to monocyte cultures, and incubated for 72 h. Changes in the number of GFP-positive cells was observed by fluorescence microscopy and compared with a mock-treated latently infected cell controls that was set to 100% (**a**). These monocytes were then differentiated using granulocyte-macrophage colony-stimulating factor (GM-CSF) and interleukin-4 (IL-4) stimulation followed by LPS treatment to mature dendritic cells (mDCs). Similarly, changes in the percentage of GFP-positive mDCs were then measured (**b**). Finally, the cells shown in **b** were co-cultured with fibroblasts for 2 weeks and the numbers of IE foci were counted (**c**). Means and error bars (showing s.d.) were generated from five independent experiments. (**d**) Monocytes were isolated and experimentally infected with either HCMV Titan wild-type or HCMV Titan with a US28 deletion at an MOI of 5. Cultures were then either mock-treated with PBS or treated with F49A-FTP for 3 days and then reactivated by differentiation into mDCs as above. Using the UL32-GFP tag on these viral isolates, reactivated dendritic cells were then counted and compared with levels of reactivation of monocytes infected with Titan wild-type virus in the absence of drug that was set to 100%. (**e**) After co-culture with reporter fibroblast cells, incubation for 2 weeks and staining for IE, reactivation events were quantified. Means and error bars (showing s.d.) were generated from three independent experiments. Statistical analyses were carried out using a paired two-tailed $t$-test and $P$ values expressed as $*P = 0.05$, $**P = 0.01$ or $***P = 0.001$.

we employed the Titan-ΔUS28 (ref. 56) that also contains a UL32-GFP tag that is expressed to high levels upon lytic infection/reactivation. We established latency in monocytes with wild-type Titan or Titan-ΔUS28, treated the latently infected cells with F49A-FTP and then reactivated cells by differentiation and maturation and quantified lytic reactivation directly by detection of UL32-GFP expression in these mDCs (Fig. 4d). We also quantified the production of infectious virus from these mDCs by co-culturing them with indicator fibroblasts and staining for viral IE reactivation foci (Fig. 4e). Although treatment of monocytes latently infected with Titan wild-type virus with F49A-FTP showed a reduction in the frequency of UL32-GFP reactivating cells (Fig. 4d) and reactivation of infectious virus from these cells (Fig. 4e), monocytes infected with Titan-ΔUS28 show no such decreases (Fig. 4d,e). This confirmed that F49A-FTP selectively kills latently infected cells based on US28 expression.

It should be pointed out that this control experiment is, necessarily, an imperfect one as US28 expression is known to be necessary for the maintenance of latency; US28-deleted viruses undergo lytic infection in undifferentiated CD34+ cells and monocytes[35]. Nevertheless, lytically infected cells are also killed by F49A-FTP when expressing US28 (see Fig. 1), and the fact that

monocytes infected with Titan-ΔUS28 are not killed by F49A-FTP still supports the notion that F49A-FTP kills latently infected monocytes in a US28-dependent manner.

In addition, we considered that killing by F49A-FTP could also be because of US28 being carried into the cell by infectious virions. In order to control for this, we delayed F49A-FTP treatment of experimentally infected monocytes by 1, 3, 5 and 7 days post infection that should allow time for virion-delivered US28 to be degraded. We found that treatment was as effective at clearing latently infected cells at day 7 compared with day 1 (Supplementary Fig. 3a). Finally, we tested F49A-FTP efficacy against cultures that were experimentally infected at low multiplicity of infection (MOI); F49A-FTP only marginally lost its efficacy when treating cultures with few latently infected monocytes (Supplementary Fig. 3b).

**F49A-FTP also kills experimentally infected CD34+ cells.** As CD34+ progenitor cells, a major component of HSCTs, are also sites of latent carriage of HCMV *in vivo*, we repeated these analyses with experimentally latently infected CD34+ progenitor cells. Firstly, we confirmed these cells were, indeed, latently infected. Consistent with previous studies[54], infected CD34+

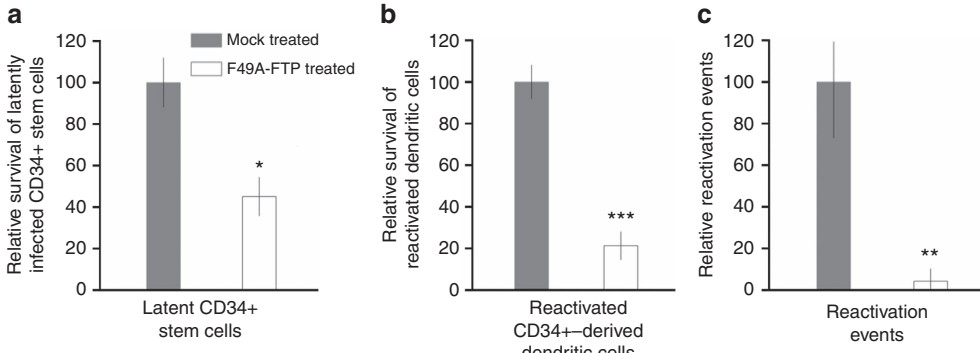

**Figure 5 | F49A-FTP kills experimentally latently infected CD34 + progenitor cells, reducing HCMV reactivation events.** CD34 + progenitor cells were experimentally infected with SV40-GFP-TB40E at an MOI of 5 that led to a mean average of 8.5% of cells latently infected. After 24 h, F49A-FTP was added to CD34 + cell cultures and incubated for 72 h. Changes in the number of GFP-positive latent cells were analysed by fluorescence microscopy and compared with a mock-treated control that was set at 100% (**a**). These CD34 + cells were then differentiated to mature dendritic cells and changes in the percentage of GFP-positive mature dendritic cells were measured (**b**). Finally, the cells shown in **b** were co-cultured with fibroblasts for 2 weeks and the number of IE foci were counted (**c**). Means and error bars (showing s.d.) were generated from three independent experiments. Statistical analyses were carried out using a paired two-tailed $t$-test and $P$ values expressed as $*P = 0.05$, $**P = 0.01$ or $***P = 0.001$.

cells showed a profile of viral transcription associated with latency; a low level of lytic IE RNA compared with much higher levels of latency-associated UL138 RNA (Supplementary Fig. 1c) and, importantly, an inability to culture infectious virus from the latently infected cells (Supplementary Fig. 1e). In contrast, differentiation and maturation of these cells to mature CD34 + derived DCs resulted in a characteristic lytic transcription profile with a switch to high levels of lytic IE compared with UL138 (Supplementary Fig. 1c), as well as efficient virus production (Supplementary Fig. 1e).

Once again, as with latently infected CD14 + monocytes, we observed that F49A-FTP reduced the number of GFP-expressing latently infected CD34 + cells (Fig. 5a), reduced the number of GFP-expressing CD34 + derived mDCs resulting from reactivation of GFP-tagged virus (Fig. 5b) and significantly reduced the amount of reactivated infectious virus capable of reinfecting HFFs in co-culture (Fig. 5c).

**F49A-FTP can kill naturally latently infected monocytes**. Our data, so far, argue that F49A-FTP is able to target and kill experimentally latently infected CD14 + monocytes and CD34 + progenitor cells, resulting in a profound reduction in virus reactivation events after differentiation and maturation of latently infected cells. However, a key question is whether such a treatment can also target naturally latent cells and prevent reactivation of infectious virus. Consequently, we isolated latently infected monocytes from healthy seropositive donors and treated them with F49A-FTP. In order to assess any reduction in naturally latent load in these cells, we differentiated and matured any surviving latently infected monocytes, after F49A-FTP treatment, to mDCs to induce virus reactivation and subsequently measured reactivation of infectious virus by co-culture with HFF indicator cells, followed by immunofluorescent staining for IE-positive foci in these fibroblast co-cultures. Figure 6 shows that treatment of naturally latently infected monocytes with F49A-FTP also causes a strong reduction in viral reactivation from these fusion toxin-treated naturally latent cells.

## Discussion
HCMV can establish latent infection in early myeloid lineage cells[4], a major cell type present in HSCTs. The presence of latent HCMV in allogenic HSCTs can lead to serious medical complications[21], accepted to be because of reactivation of latent

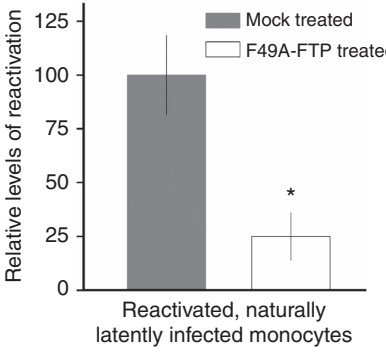

**Figure 6 | F49A-FTP kills naturally latently infected monocytes, reducing reactivation events.** $1.2 \times 10^8$ blood monocytes were isolated from CMV-positive blood donors. Half of these cells were incubated with F49A-FTP for 72 h. Monocytes were then differentiated using granulocyte-macrophage colony-stimulating factor (GM-CSF) and interleukin-4 (IL-4) stimulation followed by LPS treatment. Monocyte-derived, mature dendritic cells (reactivated latent monocytes) were then co-cultured with fibroblasts for 3 weeks, and the numbers of reactivated IE foci were detected by immunofluorescent staining. Means and error bars (showing s.d.) were generated from three independent experiments. Statistical analyses were carried out using a paired two-tailed $t$-test and $P$ values expressed as $*P = 0.05$.

cytomegalovirus after differentiation of these early myeloid cells to terminally differentiated macrophages and DCs. Currently, there are only few examples of strategies to selectively kill HCMV-infected early myeloid lineage cells[9]; this killing of latently infected cells, with the vinca alkaloid vincristine, can reduce viral reactivation events that would likely reduce their threat to any immunocompromised allo-HSCT recipient after transplant. However, the high toxicity of vincristine would likely be a serious barrier to such use therapeutically.

Recently, a rationally designed FTP named F49A-FTP has been developed that exploits cell surface expression of US28 during lytic infection to kill lytically infected fibroblast cells[46]. We have confirmed that the anti-HCMV activity of F49A-FTP is US28 specific by ectopic expression of US28 in THP-1 monocytic cells as well as demonstrating a loss of efficacy against both HFFs and monocytes that were infected with a US28-deletion isolate.

Interestingly, we and others have shown that US28 mRNA is also expressed in latently infected early myeloid lineage cells,

arguing that this HCMV-encoded, cell surface GPCR is likely expressed at the protein level during latency[34,35,49]. Using RT-qPCR and radiolabelling affinity assays, we have confirmed that US28 protein is expressed and present on the cell surface of latently infected CD14+ monocytes and this led us to ask whether this would make these latently infected cells potential targets for killing using the US28-specific fusion toxin protein, F49A-FTP. As expected, treatment of experimentally latently infected CD14+ monocytes and CD34+ progenitor cells with F49A-FTP led to a reduction in latently infected cells, as measured by decreases in GFP-expressing latent cells after infection with a GFP-tagged HCMV[50]. We also confirmed that this reduction in GFP-positive cell number was because of cell killing of GFP-expressing cells rather than suppression of GFP expression in these latently infected cells, as differentiation and maturation of these F49A-FTP-treated cells to trigger reactivation of any viable latent virus also showed a profound reduction in virus reactivation. Finally, we confirmed that naturally latent cells were also targeted by F49A-FTP by demonstrating that F49A-FTP treatment of naturally latently infected monocytes also reduced the number of reactivation events after their differentiation and maturation.

Based on these observations, we suggest that such a fusion-toxin protein treatment, targeting US28, could act as the basis for a treatment to reduce the clinical threat of HCMV reactivation after allo-HSCTs. Additionally beneficial, as infectious virus is not produced during latency, resistance to F49A-FTP because of selection of replicating, drug-resistant mutants is unlikely to occur during treatment of latently infected cells. Given that CX3CR1, a low-affinity receptor for F49A-FTP, is expressed on a number of cell types, we would not envisage treatment of patients directly with F49A-FTP. Instead, we think such a therapeutic could be developed to purge latently infected cells from grafts before engraftment; we believe this could include stem cells as well as, potentially, solid organs.

Although F49A-FTP treatment of experimentally or naturally latently infected cells did not completely ablate all reactivation events, we believe that there are a number of possible avenues to improve F49A-FTP efficacy. We chose a concentration of $5 \times 10^{-8}$ M F49A-FTP and an incubation period of 72 h as our pilot experiments in experimentally infected monocytes indicated that these were the minimum dose and time required to clear latent HCMV most efficiently while minimizing killing of uninfected monocytes (Supplementary Fig. 2). However, our analyses showed that higher concentrations of F49A-FTP could potentially be used especially in CD34+ cells that showed no signs of cell death at antivirally active concentrations of F49A-FTP (Supplementary Fig. 4). Similarly, the recent solution of the US28 crystal structure in complex with CX3CL1 (ref. 57) may allow for further refinements to F49A-FTP to increase the selectivity of F49A-FTP for US28 and therefore the efficacy of F49A-FTP against naturally latently infected tissue.

In summary, we have demonstrated that latently infected early myeloid cells express US28 that can be targeted by the FTP F49A-FTP. This results in killing of both experimentally and naturally latently infected myeloid cells and the subsequent reduction of HCMV reactivation events. These findings are proof of principle that F49A-FTP can reduce the latent load of HCMV in early myeloid cells that we believe could form the basis for an approach to reduce the clinical threat of HCMV-positive grafts in HSCTs.

## Methods

**Cell culture and viruses.** HFF cells (ATCC, 86031405) were grown in Dulbecco's modified Eagle's medium (DMEM) supplemented with 10% inactivated fetal bovine serum (PAA) and penicillin/streptomycin (Sigma). Primary CD14+ monocytes were isolated from apheresis cones (NHS blood) using Lymphoprep

(Stemcell Technologies) density gradient centrifugation, followed by magnetic-activated cell sorting using CD14+ magnetic beads (Miltenyi Biotec)[58] and cultured in X-Vivo15 (Lonza) at 37 °C in 5% $CO_2$. For experimental infection models, monocytes were incubated with virus at an MOI of 5 for 3 h at room temperature with rocking. Virus media were then removed and cells were washed twice in phosphate-buffered saline (PBS) before fresh X-Vivo15 was replaced. F49A-FTP was generated as described in Spiess et al.[46]; briefly, F49A-FTP was produced in Escherichia coli as inclusion bodies that were isolated and F49A-FTP was purified by size exclusion chromatography. After treatment with F49A-FTP, monocytes were washed and differentiated to immature dendritic cells by granulocyte-macrophage CSF and interleukin-4 (Peprotech) stimulation at 1,000 U ml$^{-1}$ for 5 days of incubation. Mature dendritic cells were produced by stimulation for 2 days with LPS (Sigma) at 500 ng ml$^{-1}$. Primary CD34+ progenitor cells were isolated from G-CSF mobilized haematopoietic stem cell donors (Charité Hospital, Berlin), in accordance with the declaration of Helsinki. These CD34+ cells were cultured in X-Vivo15 (Lonza) at 37 °C in 5% $CO_2$. CD34+ progenitor cells were differentiated to immature dendritic cells by granulocyte CSF, tumour necrosis factor TNF-α, granulocyte granulocyte-macrophage CSF and interleukin-4 (Peprotech) stimulation at 1,000 U ml$^{-1}$ for 5 days of incubation. Mature dendritic cells were produced by stimulation for 2 days with LPS at 500 ng ml$^{-1}$ (ref. 59). THP-1 cells (ATCC 8808120) were grown in RPMI-1640 (PAA), 10% inactivated fetal bovine serum (PAA) and penicillin/streptomycin (Sigma).

Titan wild type (Titan-WT) and the equivalent isolate with a deletion in the US28 gene (Titan-ΔUS28) were kind gifts from Martine Smit (Division of Medicinal Chemistry, University of Amsterdam, The Netherlands), and their generation has been previously described[60]. TB40E with an SV40-GFP tag has been described previously[14,50]. HFFs were infected at a predicted MOI of 0.1 based on titration on HFF cells. For myeloid cells, MOIs were calculated by titration on RPE-1 cells (ATCC CRL-4000). Monocytes and THP-1 were infected with all HCMV viral isolates at a predicted MOI of 5, leading to ∼10% infected cells.

For Fig. 3b, virus stock was diluted so that monocytes were infected at MOIs of 5, 2.5, 1.25, 0.625, 0.5, 0.25, 0.1 and 0.05. In order to maintain a detectable number of reactivation events, a proportional increase in the number of monocytes were infected. For the above MOIs, the number of monocytes infected were: $5 \times 10^4$, $1 \times 10^5$, $2 \times 10^5$, $4 \times 10^5$, $5 \times 10^5$, $1 \times 10^6$, $1.5 \times 10^6$ and $3 \times 10^6$, respectively.

**Treatment with F49A-FTP.** Latently infected monocytes were treated with F49A-FTP 24 h post infection, or 24 h after isolating the naturally latently infected monocytes. In Supplementary Fig. 3a, where stated, this treatment was delayed across the time course of latent infection. Based on pilot studies of titrations of F49A-FTP (Supplementary Fig. 2a), all cell cultures were treated with $5 \times 10^{-8}$ M F49A-FTP.

**Quantifying reactivation events.** Monocyte and CD34+ -derived dendritic cells were co-cultured with $3 \times 10^3$ HFFs per cm$^2$ of growth area in a 50:50 mixture of DMEM and RPMI or DMEM and X-Vivo15 (Lonza), respectively. Cultures were incubated for 2 weeks for experimental latency or 3 weeks for natural latency to allow time for reactivation events to be clearly visualized, and then stained for IE protein.

**Quantifying cell survival.** Cell death was quantified using Trypan blue staining in a 1:20 dilution in PBS. Cells were immediately observed by light microscopy and both white (live) and blue (dead) cells were counted. Cell survival was calculated as the number of white cells as a percentage of total cells.

**Immunofluorescent staining for IE protein-positive cells.** Cells were fixed and stained as described previously[61]. In brief, following fixation in paraformaldehyde and permeabilization in 70% ethanol, cells were washed in PBS and stained with mouse anti-IE (Argene 11-003) diluted in PBS (1 in 1,000) followed by goat anti-mouse (Alexafluor 488) with Hoechst nuclear stain. Cell percentages were calculated as the percentage of Hoechst-stained nuclei colocalizing with IE-positive nuclei.

**Lentivirus transduction of THP-1 cells.** A US28 construct with an N-terminal HA tag was kindly provided by Daniel Streblow (Oregan Health and Science University). This construct was subcloned into the lentiviral vector (pHR-SIN 1253) that was provided by Paul Lehner (Cambridge Institute for Medical Research) and was co-transfected with packaging and envelope plasmids into 293T cells (ECACC 85120602). Lentivirus was harvested after 48 and 72 h and incubated with THP-1 cells in the presence of polybrene for transduction of US28 expression.

**Detection of US28 expression by immunoblot.** THP-1 cells, transduced to express HA-US28, were lysed in RIPA buffer (supplemented with 0.1% SDS, 2 mM EDTA and 1 mM phenylmethylsulfonyl fluoride). Nuclei and cell debris were removed by centrifugation at 13,000 × g for 10 min at 4 °C. Proteins were separated on 10% SDS-polyacrylamide gels and transferred to nitrocellulose membranes

(Axygen, Corning). Incubations with primary and secondary antibodies were in 5% skimmed milk for 1 h each at room temperature. To detect HA-US28, mouse anti-HA antibody (1 μg ml$^{-1}$) (F-7 Santa Cruz Biotechnology) was used followed by bovine anti-mouse horseradish peroxidase (Santa Cruz Biotech). Blots were developed with the use of enhanced chemiluminescence (GE Healthcare) and visualized with autoradiography film. A full version of the blot in Fig. 2 can be seen in Supplementary Fig. 5.

**Homologous binding experiments on transfected COS-7 cells.** $3 \times 10^6$ COS-7 cells (ATCC CRL-1651) were transfected with 20 μg of receptor cDNA (US28 or CX3CR1) using the calcium precipitation method[56] and transferred to 24-well culture plates coated with poly-D-lysine 1 day after transfection. The number of cells seeded per well was determined by the apparent expression efficiency of the receptors and was aimed at obtaining 5–10% specific binding of the added radioactive ligand ($2.5 \times 10^4$ cells per well for US28 and $1 \times 10^5$ cells per well for CX3CR1-expressing cells). At 2 days after transfection, cells were assayed by competition binding for 3 h at 4 °C using 10–15 M $^{125}$I-CX3CL1 as well as unlabelled CX3CL1 in 50 mM HEPES buffer, pH 7.4, supplemented with 1 mM CaCl$_2$, 5 mM MgCl$_2$ and 0.5 (w/v) bovine serum albumin (binding buffer)[62]. After incubation, cells were washed twice in ice-cold binding buffer, supplemented with 0.5 M NaCl. Experiments were performed in duplicate.

**Homologous binding experiments on monocytes.** Approximately $2 \times 10^8$ monocytes were infected with SV40-GFP-TB40E at an MOI of 5. At 4 days after infection, monocytes were sorted for GFP-positive cells, yielding ∼5% of total cells. Approximately $7 \times 10^4$ of these GFP-positive monocytes were seeded per well to 96-well culture plates coated with poly-D-lysine. At 3 h after seeding, cells were assayed by competition binding as described for COS-7 cells above. Determinations were made in duplicate.

**RT-qPCR analysis of viral and cellular gene expression.** For RT-qPCR, RNA was harvested using TRIzol reagent (Life Technologies) and isolated using RNeasy mini kit (Qiagen), following the manufacturer's instructions. UL138, IE and UL99 were quantified using a one-step RT-qPCR using Quantitect Virus kit (Qiagen) as previously described[12]. UL138 was detected using the following primers and probe: 5′-CTGGATACACGGTACACA-3′ (forward), 5′-TACGGGAATTCAGTGGAT A-3′ (reverse), and [cy5] 5′-CCTGCGACGACTACACTATGAATACA-3′ [BHQ2] (probe). IE was detected using the following primers and probe: 5′-GGGTA TGTGTCTGAAAATGA-3′ (forward), 5′-GACTCTTATCAGAACACAACAA-3′ (reverse), and [6FAM] 5′-ATCTTCTTCTGCCGCTGCCTTA-3′ [BHQ2] (probe). UL99 was detected using the following primers and probe: 5′-CGAACTCTGCA AACGAATA-3′ (forward), 5′-GAGGGATGTTGTCGTAGG-3′ (reverse), and [cy3]5′-CGTAGAGACACCTGGCGACC-3′ [BHQ2] (probe). Samples were normalized to glyceraldehyde 3-phosphate dehydrogenase (GAPDH), as described previously[12], using the following primers and probe: 5′-CCCTTCATTGACCTC AAC-3′ (forward), 5′-GGTGATGGGATTTCCATTG-3′ (reverse), and [JOE] 5′-CTCAGCCTTGACGGTGCCAT-3′ [BHQ2].

CX3CR1 and US28 were detected using Quantitect SYBR Green RT-PCR Kit (Qiagen), following the manufacturer's instructions, and the following primers: 5′-CTGCCTCTTAGACTTCTG-3′ (forward), 5′-GGCTATCACTCTGTAGAC-3′ (reverse). US28 was detected using the following primers: for the forward primer: 5′-TCGCGCCACAAAGGTCGCATTG-3′, and for the reverse primer: 5′-CGACACACCTCGTCGGACAGCG-3′. Values were calculated using the ΔΔCT method relative either to uninfected monocytes (for CX3CR1) or monocytes immediately after infection with HCMV (for US28). Samples were normalized to GAPDH, as described previously[12].

**Ethics statement.** All human samples were obtained under ethical approval and after approval of protocols from the Cambridgeshire 2 Research Ethics Committee (REC reference 97/092) conducted in accordance with the Declaration of Helsinki. Informed written consent was obtained from all of the volunteers included in this study before providing blood samples and all experiments were carried out in accordance with the approved guidelines.

**Data availability.** All relevant data are available from the authors on request.

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

## Acknowledgements

We thank Linda Teague (Department of Medicine, Cambridge, UK), Olav Larsen and Maibritt Baggesen (Laboratory for Molecular Pharmacology, Department of Neuroscience and Pharmacology, University of Copenhagen, Denmark) for technical assistance. We thank M. Smit (Division of Medicinal Chemistry, University of Amsterdam, The Netherlands) for providing the Titan HCMV wild type and Titan-ΔUS28 viral isolates, D. Streblow (Oregan Health and Science University, USA) for the US28 gene and P. Lehner (CIMR, Cambridge, UK) for the lentivirus vector. This work was funded by the British Medical Research programme Grant G0701279 (to J.H.S.), Wellcome Research Studentship Grant (to B.A.K.), the Hørslev Foundation (to M.M.R.) and the Cambridge NIHR BRC Cell Phenotyping Hub.

## Author contributions

B.A.K., E.L.P. and K.S. performed experiments and analysed data. S.V. provided reagents. K.S., T.N.K., M.M.R., E.L.P. and J.H.S. designed experiments. B.A.K. and J.H.S. wrote the manuscript. All authors read and edited the manuscript.

## Additional information

**Competing financial interests:** The authors declare no competing financial interests.

**Publisher's note**: 

