## [Peer Review File · Nature Communications]

Reviewers' comments:

Reviewer #1 (Remarks to the Author):

A. Summary of key results: In the article entitled, "Targeting the latent cytomegalovirus reservoir with a novel fusion toxin protein" by B.A. Krishna, et al., the authors have shown that they can specifically target human cytomegalovirus (HCMV) latently infected cells with a fusion toxin protein (F49A-FTP), which thereby reduces reactivation in these cells in response to stimuli. The authors have shown that F49A-FTP targets and kills these infected cells by interaction with the viral G-protein coupled receptor (GPCR), US28. US28 is one of the HCMV latently expressed proteins, which the authors show in this manuscript is expressed on the cell surface of the latently infected cells. The authors suggest that this novel targeting of latently infected cell could provide a platform for novel therapeutic strategies for purging HCMV latently infected cells prior to hematopoietic stem cell transplantation.

B. Originality and interest: The findings the authors present in this manuscript are novel, as they represent a new means by which latently infected myeloid cells could effectively be removed from the hematopoietic system prior to hematopoietic stem cell transplantation.

C. Data & methodology: The validity of the approach, the quality of the data, and the quality of the data presentation are acceptable.

D. Appropriate use of statistics and treatment of uncertainties: The data presented represent findings from multiple independent biological replicates, and the statistics have been presented appropriately.

E. Conclusions: Overall, the conclusions the authors have drawn are mostly supported by their data. Several improvements (listed below in Section F) would help to bolster their conclusions.

F. Suggested Improvements: Several points, bulleted below, if addressed, would strengthen the manuscript.

a. Some of the methodologies used in the experiments can be better clarified.

i. How many days were the cells latently infected prior to treatment with F49A-FTP? It seems like the myeloid cells were cultured for 5 days prior to treatment (lines 283 and 287) or 24h (line 644, Figure 5 figure legend). It is not overly clear which of these time points was used as the duration of latency prior to treatment, since there is no description of the toxin treatment in the methods.

ii. Are the number of myeloid cells that are co-cultured with fibroblasts equal, or are they the total cell cultures that were originally infected then treated +/- F49A-FTP? It seems equal myeloid cell numbers should be used in the co-culture experiments, because if more cells are dying in the F49A-FTP, then the reactivation results could be skewed just due to the actual number of cells in the untreated (F49A-FTP minus) cultures. Also, if equal cell number were used, how do the authors rid the cultures of the dead cells prior to co-culture?

iii. Are the RT-qPCR data normalized to a cellular gene? If not, how are these data normalized across samples and conditions?

iv. In the figure legend for Figures 5 and S2, the authors state that the myeloid cells were co-cultured with fibroblasts for 2 weeks, however the methods and elsewhere (line 659, Figure 6 figure legend) says 3 weeks (or 21 days). Which is correct?

b. The data in Figure 3 does not directly show that US28 is localized to the cell surface of the infected monocytes (nor do the authors overstep to make this claim). Additionally, this reviewer appreciates the difficulty in directly showing this without a suitable antibody that recognizes US28 or an epitope tag fused to US28 to allow for detection of this viral protein (though several HCMV clinical isolates with a US28 epitope tag have been published). However, can the authors be sure that HCMV infection alone does not upregulate CX3CR1 on the monocytes? To ensure that this finding is a direct result of CX3CL1 binding to US28, the authors could include US28 Δ infected monocytes, which would likely follow the trend of the uninfected monocytes if HCMV infection alone does not cause CX3CR1 upregulation. Additionally, the authors could perform RT-qPCR for CX3CR1 in wild type and US28 Δ -infected monocytes compared to uninfected monocytes to show that the endogenous receptor is not upregulated by HCMV infection alone.

G. References: The manuscript is appropriately referenced, though there are two references for the TB40 GFP virus - reference numbers 50 (lines 143-144) and 14 (line 295).

Presumably this is the same virus, thus it seems reference 50 is appropriate on line 295.

H. Clarity and context: This is a well-written manuscript. The abstract provides appropriate information and is clear in its content. The introduction and conclusions are both clear - the introduction puts the study into context and the discussion summarizes the salient points of the manuscript and discusses the implication of the findings.

Reviewer #2 (Remarks to the Author):

The goal of this work is to show that myeloid cells (CD14+ and CD34+ cells) latently infected with human cytomegalovirus (CMV) express the chemokine receptor US28, and can therefore be eliminated by exposure to the fusion toxin protein and US28 ligand F49A-FTP. The biggest novelty of the current data lies on the introduction of a new target (a viral protein rather than a cellular one) and of a new killing agent, which may provide a less toxic therapeutic alternative than vincristine, as previously discussed in a beautiful work recently published by Dr. Sinclair in Science (Weekes MP, et al., Science, 2013). In this reviewer's opinion, however, a number of modifications and additional data should be gathered to render data fully convincing:

Figure 1. Why are there so many infected cells for an MOI of 0.1 (less than 10% should be expected)? Why is the number of Titan-deltaUS28 infected cells so much lower than that of wild-type infected cells? Images of Hoechst stained cell nuclei should be shown to prove that F49A-FTP is selective for infected cells expressing US28. As less than 10% of the cells should be infected at an MOI of 0.1, there should still be plenty of Hoechst+ cells in the F49A-FTP treated, Titan-deltaUS28 infected cells. This is provided that the drug does not, indeed, kill uninfected cells.

Figure S1. There's only two panels (A and B), and no demonstration "that F49A-FTP does not trigger virus reactivation" as stated in the legend. A panel and the corresponding data seem to be missing. Although popularly used as a "marker" of latent infections, the UL138 gene is also abundantly expressed during lytic replication. Its presence per se is thus not a

good indicator of true latency. As infected monocytes and, even more so, CD34+ cells seem to still contain large amounts of both IE and UL99 transcripts, further proof of real latency is required. Expression of UL99, a true late gene, is particularly worrisome.

Immunofluorescence staining analyses of infected cells using anti-UL99 Ab would help determine if all cells express this protein (indicating that infection is lytic or, at least, persistent) or if only a subset of cells do (suggesting that infection is latent in the majority of the cells, but lytic in a sub-population). Also, use of additional latency markers as well as titrating of intracellular and extracellular virus amounts present in infected cells at different times post direct infection (not after co-culture with HFF) would help determine how "tight" latency actually is in these models. This is extremely important, as different models using different cytokine mixes and culture conditions can lead to dramatically diverse results, although this is not yet fully recognized in the pertinent literature.

Figure 3. At what time post-infection were these assays conducted? How do we know that the binding of 125iodine-radiolabelled CX3CL1 on infected cells is due to US28 and not to the virus-induced upregulation of CX3CR1? This experiment should be repeated using the Titan-WT and the Titan- Δ US28 viruses. Although these viruses do not express GFP and are not amenable to FACS sorting, the difference between the two infected populations should be readily detectable. Also, real-time qPCR analyses should be conducted to quantify the expression levels of both US28 and CX3CL1 at different times during infection. Although this assay will not discern if these molecules are on the surface or not, it will help determine if expression of these genes is upregulated by infection.

Figure 4 and 5. US28 expression has been detected quite late post-infection of myeloid cells (day 8 in CD34+ cells, Goodrum, 2002, day 3 in GM-PS, Cheung, 2006, day 6 in monocytes, Poole, 2013). The only report of US28 transcripts being present at earlier times is from Beisser et al., 2001, and describes RNA molecules introduced with the virions upon entry. This work was conducted with THP-1 cells, not monocytes, and did not disclose whether the US28 protein was expressed and displayed on the cell surface. Therefore, addition of F49A-FTP at 24 hours post monocyte infection and its maintenance until day 3 is not very likely to have actually targeted US28. Rather, it may have acted on CX3CR1. As above, to be fully convincing, these assays should be redone using Titan and Titan- Δ US28 infected cells, and evidence independent of GFP expression from the SV40 cassette (whose reliability is limited being that it may be affected by factors that have nothing to do with viral replication or latency) should be gathered.

Figure 6. This is the best and most pristine evidence that F49A-FTP can indeed target and kill latently infected cells, although of course it does not show that this is via binding to US28 (as opposed, for instance, to upregulated CXCR1). Nevertheless, in my opinion, this is the best figure of the paper, and would gain even more power should a similar killing of CD34+ cells be shown.

Reproducibility:

The Titan strain and its derivative, Titan- Δ US28 have not been previously described (and thus should be in this work). How was this new strain produced? How was the delta-US28 made?

How was the SV40-GFP tagged virus derived? Where was the SV40-GFP tag inserted? TB40/E is a genetically heterogeneous strain originally isolated by C. Sinzger. As such, it cannot have contained any markers. Consequently, the SV40-GFP tagged version of this virus must have been derived from a clone (likely a BAC). BAC insertion as well as any other genomic modifications entails the loss of specific viral genes. The resulting virus, therefore, is not identical to the parental strain from which it was derived, and should be addressed as such. How was the SV40-GFP strain derived? Was this produced starting from TB40-BAC4 (a clone of TB40/E)? If so, this strain must be properly named in the paper, as it is not TB40/E.

Clarity and context:

A few sentences need revision.

In these latently infected cells, maintenance (maintenance) of viral genome is associated with expression of a relatively small number of latency-associated viral genes and no infectious virus is produced by latently infected cells (redundant).

These changes are likely mediated by expression of the HCMV latency-associated genes (15,16), a . A number of which have been detected in naturally latently infected cells from healthy seropositive donors; these include: UL84, UL81-82ast, UL138, UL144, LUNA, UL111A, Lnc4.9 and US28 (16-20).

However, this differentiation can also result in HCMV reactivation in up to 80% of allo-HSCT patients, if left untreated, without anti-virals (21). Left untreated and without antivirals may be the same thing?

Latent monocytes have high affinity for CX3CL1, indicating cell surface US28 expression. Perhaps it is meant "Latently-infected monocytes"?

Reviewer #3 (Remarks to the Author):

In this manuscript Krishna and colleagues have explored a novel potential strategy to target human cytomegalovirus (HCMV) latently infected cells through a cell surface G protein-coupled receptor US28 encoded by this virus. They have used a fusion toxin protein (F49A-FTP) which is based on the soluble extracellular domain of the US28 ligand CX3CL1 and a recombinant *Pseudomonas* exotoxin-A motif. This toxin binds to US28 with high selectivity compared to the cellular CX3CL1 receptor, CX3CR1. After binding US28, F49A-FTP is internalised and mediates cell killing via the bacterial toxin domain. Authors demonstrate through a series of experiments based on lytically infected fibroblasts, latently infected myeloid (including monocytes) cells, US28 expressing THP-1 cells and CD34+ cells that F49A-FTP is highly effective in killing all these different cell types. More importantly, F49A-FTP treatment also reduced HCMV reactivation event. Overall this is an interesting study and extends the original paper published by Spiess and colleagues (PNAS; July 7, 2015; vol. 112; no. 27; pp8427-8432) which also showed that F49A-FTP can block replicative HCMV infection in vitro and in vivo. Most the data is highly convincing and well presented.

While the extension of the original observations published in PNAS to control of latent infection is indeed a step forward, I am not sure if this is really a major advance especially in the context of the current format of the data. I have series of concerns which authors need to address before this manuscript can be considered suitable for publication.

(a) Much of the data in this manuscript is based on a single F49A-FTP dose (no information on the concentration of protein in figure legends or results section) and fixed time point analysis. It is absolutely essential authors carry out a more detailed dose response and time course analysis. It is very difficult to judge the therapeutic value of F49A-FTP especially for clinical setting without this detailed analysis.

(b) Similarly, in all experiments single viral infection dose has been used for therapeutic assessment. It will be important to see if F49A-FTP treatment can control varying levels of HCMV infection.

(c) None of the figures have statistical analysis; especially figure 3.

(d) I was wondering if authors have considered potential bystander effects of F49A-FTP on non-myeloid cells. F49A-FTP can bind CX3CR1 which is also expressed in differentiated human ciliated airway cells. Does this mean this protein may have some serious clinical problem for humans especially in stem cell transplant patients who are severely immunosuppressed? Authors should discuss this potential risk of F49A-FTP.

Answers to Reviewers' comments:

Reviewer #1 (Remarks to the Author):

A. Summary of key results: In the article entitled, "Targeting the latent cytomegalovirus reservoir with a novel fusion toxin protein" by B.A. Krishna, et al., the authors have shown that they can specifically target human cytomegalovirus (HCMV) latently infected cells with a fusion toxin protein (F49A-FTP), which thereby reduces reactivation in these cells in response to stimuli. The authors have shown that F49A-FTP targets and kills these infected cells by interaction with the viral G-protein coupled receptor (GPCR), US28. US28 is one of the HCMV latently expressed proteins, which the authors show in this manuscript is expressed on the cell surface of the latently infected cells. The authors suggest that this novel targeting of latently infected cell could provide a platform for novel therapeutic strategies for purging HCMV latently infected cells prior to hematopoietic stem cell transplantation.

B. Originality and interest: The findings the authors present in this manuscript are novel, as they represent a new means by which latently infected myeloid cells could effectively be removed from the hematopoietic system prior to hematopoietic stem cell transplantation.

C. Data & methodology: The validity of the approach, the quality of the data, and the quality of the data presentation are acceptable.

D. Appropriate use of statistics and treatment of uncertainties: The data presented represent findings from multiple independent biological replicates, and the statistics have been presented appropriately.

E. Conclusions: Overall, the conclusions the authors have drawn are mostly supported by their data. Several improvements (listed below in Section F) would help to bolster their conclusions.

F. Suggested Improvements: Several points, bulleted below, if addressed, would strengthen the manuscript.

a. Some of the methodologies used in the experiments can be better clarified.

i. How many days were the cells latently infected prior to treatment with F49A-FTP? It seems like the myeloid cells were cultured for 5 days prior to treatment (lines 283 and 287) or 24h (line 644, Figure 5 figure legend). It is not overly clear which of these time points was used as the duration of latency prior to treatment, since there is no description of the toxin treatment in the methods.

We apologise for this lack of detail. We have, now, improved the description of F49A-FTP treatment in the methods section, which clarifies this misunderstanding.

ii. Are the number of myeloid cells that are co-cultured with fibroblasts equal, or are they the total cell cultures that were originally infected then treated +/- F49A-FTP? It seems equal myeloid cell numbers should be used in the co-culture experiments, because if more cells are dying in the F49A-FTP, then the reactivation results could be skewed just due to the actual number of cells in the untreated (F49A-FTP minus) cultures. Also, if equal cell number were used, how do the authors rid the cultures of the dead cells prior to co-culture?

At the concentration of F49A-FTP that we used, we saw around 20% cell death in monocytes, and negligible levels of CD34+ cell death (Figure S4). Consequently, when we differentiated these cells, we saw little difference in the density of monocyte or CD34+-derived dendritic cells. Additionally, we would expect a reduction in monocyte survival of 20% to result in a reduction in reactivation events of 20%, however we see a near-complete reduction in events. Although we cannot rule out some contribution of F49A-FTP to be due to general cell death, it clearly shows very strong specificity. To address this issue, we have also included a partial control, treating monocytes infected with Titan-ΔUS28 and demonstrating that F49A-FTP is unable to kill these cells specifically.

iii. Are the RT-qPCR data normalized to a cellular gene? If not, how are these data normalized across samples and conditions?

In the figure legend, we now clarify that data was normalised to GAPDH.

iv. In the figure legend for Figures 5 and S2, the authors state that the myeloid cells were co-cultured with fibroblasts for 2 weeks, however the methods and elsewhere (line 659, Figure 6 figure legend) says 3 weeks (or 21 days). Which is correct?

For experimental latency, we co-cultured the early myeloid cell-derived dendritic cells for two weeks with fibroblasts and could clearly see IE-positive foci. However, for natural latency, we found that incubation for three weeks was necessary to properly visualise foci. We have clarified this difference in the methods section.

b. The data in Figure 3 does not directly show that US28 is localized to the cell surface of the infected monocytes (nor do the authors overstep to make this claim). Additionally, this reviewer appreciates the difficulty in directly showing this without a suitable antibody that recognizes US28 or an epitope tag fused to US28 to allow for detection of this viral protein (though several HCMV clinical isolates with a US28 epitope tag have been published). However, can the authors be sure that HCMV infection alone does not upregulate CX3CR1 on the monocytes? To ensure that this finding is a direct result of CX3CL1 binding to US28, the authors could include US28 Δ infected monocytes, which would likely follow the trend of the uninfected monocytes if HCMV infection alone does not cause CX3CR1 upregulation. Additionally, the authors could perform RT-qPCR for CX3CR1 in wild type and US28 Δ -infected monocytes compared to uninfected monocytes to show that the endogenous receptor is not upregulated by HCMV infection alone.

As suggested, we have performed a RT-qPCR for CX3CR1 at three time points (day 1, 3 and 5 post latent infection) and show that CX3CR1 expression does not change. We also performed an RT-qPCR to detect US28 to show that it is expressed from one day PI. We have also included, as requested, RT-qPCR for CX3CR1 in Δ US28 infected monocytes.

G. References: The manuscript is appropriately referenced, though there are two references for the TB40 GFP virus - reference numbers 50 (lines 143-144) and 14 (line 295). Presumably this is the same virus, thus it seems reference 50 is appropriate on line 295.

Apologies, we have now corrected this on line 295.

H. Clarity and context: This is a well-written manuscript. The abstract provides appropriate information and is clear in its content. The introduction and conclusions are both clear - the introduction puts the study into context and the discussion summarizes the salient points of the manuscript and discusses the implication of the findings.

Reviewer #2 (Remarks to the Author):

The goal of this work is to show that myeloid cells (CD14+ and CD34+ cells) latently infected with human cytomegalovirus (CMV) express the chemokine receptor US28, and can therefore be eliminated by exposure to the fusion toxin protein and US28 ligand F49A-FTP. The biggest novelty of the current data lies on the introduction of a new target (a viral protein rather than a cellular one) and of a new killing agent, which may provide a less toxic therapeutic alternative than vincristine, as previously discussed in a beautiful work recently published by Dr. Sinclair in Science (Weekes MP, et al., Science, 2013). In this reviewer's opinion, however, a number of modifications and additional data should be gathered to render data fully convincing:

Figure 1. Why are there so many infected cells for an MOI of 0.1 (less than 10% should be expected)? Why is the number of Titan-deltaUS28 infected cells so much lower than that of wild-type infected cells? Images of Hoechst stained cell nuclei should be shown to prove that F49A-FTP is selective for infected cells expressing US28. As less than 10% of the cells should be infected at an MOI of 0.1, there should still be plenty of Hoechst+ cells in the F49A-FTP treated, Titan-deltaUS28 infected cells. This is provided that the drug does not, indeed, kill uninfected cells.

We apologise for this. Indeed, the original experiment was not an MOI of 0.1, as correctly surmised by the reviewer but an MOI of 1. We have repeated this experiment at an MOI of 0.1 and included a Hoechst stain, as suggested by the reviewer.

Figure S1. There's only two panels (A and B), and no demonstration "that F49A-FTP does not trigger virus reactivation" as stated in the legend. A panel and the corresponding data seem to be missing. Although popularly used as a "marker" of latent infections, the UL138 gene is also abundantly expressed during lytic replication. Its presence per se is thus not a good indicator of true latency. As infected monocytes and, even more so, CD34+ cells seem to still contain large amounts of both IE and UL99 transcripts, further proof of real latency is required. Expression of UL99, a true late gene, is particularly worrisome. Immunofluorescence staining analyses of infected cells using anti-UL99 Ab would help determine if all cells express this protein (indicating that infection is lytic or, at least, persistent) or if only a subset of cells do (suggesting that infection is latent in the majority of the cells, but lytic in a sub-population). Also, use of additional latency markers as well as titrating of intracellular and extracellular virus amounts present in infected cells at different times post direct infection (not after co-culture with HFF) would help determine how "tight" latency actually is in these models. This is extremely important, as different models using different cytokine mixes and culture conditions can lead to dramatically diverse results, although this is not yet fully recognized in the pertinent literature.

We apologise that figure S1 appears to have been cropped, which we have now fixed. To address the question of whether our analysis demonstrates true latency, we have stained our infected monocytes with antibodies against both UL99 and immediate early, as suggested by the reviewer, and do not see expression of these proteins. However, we do see this expression in lytically infected dendritic cells. We are also unable to detect release of virus from monocytes after co-culture with fibroblasts, which is, perhaps, the gold standard test.

Figure 3. At what time post-infection were these assays conducted? How do we know that the binding of 125iodine-radiolabelled CX3CL1 on infected cells is due to US28 and not to the virus-induced upregulation of CX3CR1? This experiment should be repeated using the Titan-WT and the Titan-ΔUS28 viruses. Although these viruses do not express GFP and are not amenable to FACS sorting, the difference between the two infected populations should be readily detectable. Also, real-time qPCR analyses should be conducted to quantify the expression levels of both US28 and CX3CL1 at different times during infection. Although this assay will not discern if these molecules are on the surface or not,

it will help determine if expression of these genes is upregulated by infection.

We have clarified that these analyses were performed 5 days post-infection. We have also, as requested, performed RT-qPCR for CX3CR1 at three time points (day 1, 3 and 5 post infection) and show that expression does not change due to infection after day 3. Attempting to perform this analysis with the Titan-WT and Titan Δ -US28, without FACS sorting, would likely be too insensitive due to the large amount of CX3CR1 on the uninfected cells in each culture (at least 90% of all cells). Additionally, as it appears that US28 is necessary for the establishment of latency (Humby and O'Connor 2015 and our unpublished observations), Titan Δ -US28 would be an unsuitable control as all other viral lytic genes will be expressed in a Δ US28 background. Instead, we have included data whereby we treated Δ US28 infected monocytes with F49A-FTP, and shown that F49A-FTP is no longer effective. This indicates that the efficacy of F49A-FTP is due to US28 expression and not changes in CX3CR1.

Figure 4 and 5. US28 expression has been detected quite late post-infection of myeloid cells (day 8 in CD34+ cells, Goodrum, 2002, day 3 in GM-PS, Cheung, 2006, day 6 in monocytes, Poole, 2013). The only report of US28 transcripts being present at earlier times is from Beisser et al., 2001, and describes RNA molecules introduced with the virions upon entry. This work was conducted with THP-1 cells, not monocytes, and did not disclose whether the US28 protein was expressed and displayed on the cell surface. Therefore, addition of F49A-FTP at 24 hours post monocyte infection and its maintenance until day 3 is not very likely to have actually targeted US28. Rather, it may have acted on CX3CR1. As above, to be fully convincing, these assays should be redone using Titan and Titan-deltaUS28 infected cells, and evidence independent of GFP expression from the SV40 cassette (whose reliability is limited being that it may be affected by factors that have nothing to do with viral replication or latency) should be gathered.

We have performed RT-qPCR analysis to show that US28 expression occurs at early time points post infection, in our model of latency (figure 3D). As stated above, we have also included data from monocytes infected with Δ US28, to demonstrate that US28 expression is indeed necessary for targeting with F49A-FTP.

Figure 6. This is the best and most pristine evidence that F49A-FTP can indeed target and kill latently infected cells, although of course it does not show that this is via binding to US28 (as opposed, for instance, to upregulated CXCR1). Nevertheless, in my opinion, this is the best figure of the paper, and would gain even more power should a similar killing of CD34+ cells be shown.

The data that we have included in this revised manuscript (mentioned above) now shows that changes in CX3CR1 expression are not contributing to the efficacy of F49A-FTP.

We do agree that being able to show killing of naturally latent CD34+ cells would be powerful. Unfortunately, despite our attempts over the last 6 weeks, we have been unable to procure residual CD34+ cells from an HCMV positive CD34+ stem cell donation. However, we have clearly shown that F49A-FTP kills both experimentally and naturally latent CD14+ monocytes and we have also clearly demonstrated that F49A-FTP can kill experimentally latently infected CD34+ cells, which we believe is strong evidence for the ability of this novel therapeutic to target latently infected cells.

Reproducibility:

The Titan strain and its derivative, Titan- Δ US28 have not been previously described (and thus should be in this work). How was this new strain produced? How was the delta-US28 made?

We apologise for not making it clear that Titan- Δ US28 has been previously described by Maussang et al PNAS 2006. We have added this reference in the manuscript.

How was the SV40-GFP tagged virus derived? Where was the SV40-GFP tag inserted?
TB40/E is a genetically heterogeneous strain originally isolated by C. Sinzger. As such, it cannot have contained any markers. Consequently, the SV40-GFP tagged version of this virus must have been

derived from a clone (likely a BAC). BAC insertion as well as any other genomic modifications entails the loss of specific viral genes. The resulting virus, therefore, is not identical to the parental strain from which it was derived, and should be addressed as such. How was the SV40-GFP strain derived? Was this produced starting from TB40-BAC4 (a clone of TB40/E)? If so, this strain must be properly named in the paper, as it is not TB40/E.

We agree with the reviewer's points and we have clarified the derivation of the viruses we have used. The derivation of the SV40-GFP tagged isolate of TB40e, which is, indeed BAC derived, has been described previously (which is cited in the manuscript as O'Connor and Murphy J Virol 2013). We have now referred to this virus as "SV40-GFP-TB40E", which is the same nomenclature as has been used previously (Mason et al PNAS 2012).

Clarity and context:

A few sentences need revision.

In these latently infected cells, maintenance (maintenance) of viral genome is associated with expression of a relatively small number of latency-associated viral genes and no infectious virus is produced by latently infected cells (redundant).

These changes are likely mediated by expression of the HCMV latency-associated genes (15,16), a number of which have been detected in naturally latently infected cells from healthy seropositive donors; these include: UL84, UL81-82ast, UL138, UL144, LUNA, UL111A, Lnc4.9 and US28 (16-20).

However, this differentiation can also result in HCMV reactivation in up to 80% of allo-HSCT patients, if left untreated, without anti-virals (21). Left untreated and without antivirals may be the same thing?

Latent monocytes have high affinity for CX3CL1, indicating cell surface US28 expression. Perhaps it is meant "Latently-infected monocytes"?

These have all been corrected, as requested.

Reviewer #3 (Remarks to the Author):

In this manuscript Krishna and colleagues have explored a novel potential strategy to target human cytomegalovirus (HCMV) latently infected cells through a cell surface G protein-coupled receptor US28 encoded by this virus. They have used a fusion toxin protein (F49A-FTP) which is based on the soluble extracellular domain of the US28 ligand CX3CL1 and a recombinant Pseudomonas exotoxin-A motif. This toxin binds to US28 with high selectivity compared to the cellular CX3CL1 receptor, CX3CR1. After binding US28, F49A-FTP is internalised and mediates cell killing via the bacterial toxin domain. Authors demonstrate through a series of experiments based on lytically infected fibroblasts, latently infected myeloid (including monocytes) cells, US28 expressing THP-1 cells and CD34+ cells that F49A-FTP is highly effective in killing all these different cell types. More importantly, F49A-FTP treatment also reduced HCMV reactivation event. Overall this is an interesting study and extends the original paper published by Spiess and colleagues (PNAS; July 7, 2015; vol. 112; no. 27; pp8427-8432) which also showed that F49A-FTP can block replicative HCMV infection in vitro and in vivo. Most the data is highly convincing and well presented. While the extension of the original observations published in PNAS to control of latent infection is indeed a step forward, I am not sure if this is really a major advance especially in the context of the current format of the data. I have series of concerns which authors need to address before this manuscript can be considered suitable for publication.

(a) Much of the data in this manuscript is based on a single F49A-FTP dose (no information on the concentration of protein in figure legends or results section) and fixed time point analysis. It is absolutely essential authors carry out a more detailed dose response and time course analysis. It is very difficult to judge the therapeutic value of F49A-FTP especially for clinical setting without this detailed analysis.

Figure S2 originally included a dose analysis, but now also includes a time point analysis. We mention in the manuscript that these analyses were used as the basis for our further work.

(b) Similarly, in all experiments single viral infection dose has been used for therapeutic assessment. It will be important to see if F49A-FTP treatment can control varying levels of HCMV infection.

We thank the reviewer for this suggestion. Figure S3 now includes an analysis of changing viral infection doses.

(c) None of the figures have statistical analysis; especially figure 3.

Statistical analysis has now been included, as requested.

(d) I was wondering if authors have considered potential bystander effects of F49A-FTP on non-myeloid cells. F49A-FTP can bind CX3CR1 which is also expressed in differentiated human ciliated airway cells. Does this mean this protein may have some serious clinical problem for humans especially in stem cell transplant patients who are severely immunosuppressed? Authors should discuss this potential risk of F49A-FTP.

We have considered this possibility, and for this reason we only propose that F49A-FTP be used to treat grafted cells and not whole patients. This has been clarified in the discussion.

Reviewers' comments:

Reviewer #1 (Remarks to the Author):

In this resubmission by B.A. Krishna, et al, entitled "Targeting the latent cytomegalovirus reservoir with a novel fusion toxin protein", the authors show that F49A-FTP specifically targets latently infected hematopoietic cells (CD14+, CD34+, and THP-1) via interaction with US28. This thereby reduces the amount of cells capable of reactivation in response to the appropriate stimuli. As the authors suggest, this could prove a novel mechanism by which to rid (or drastically reduce) a bone marrow graft of HCMV latently infected cells prior to hematopoietic transplantation, thus reducing the possibility for cytomegalovirus reactivation and increasing quality of life for the transplant recipient. The authors have made significant changes to the manuscript from the first review. Importantly, they have alleviated many of the concerns raised by the three reviewers by including some additional experiments as well as clarifying sections of the text. The experiments that the authors added (e.g.: time course RTqPCR data for CX3CR1, Δ US28-infected monocytes +/- F49A-FTP, US28 RTqPCR time course data, added time course analyses of F49A-FTP, effectiveness of F49A-FTP over various MOI's) have significantly improved the manuscript from the first submission. The clarity added to the methods, results, and discussion sections have also greatly improved the manuscript on a whole. To the point of Reviewer #2 regarding the TB40/E variant used herein, it seems appropriate to use the abbreviation that was published in the original manuscript to avoid confusion, as there are an increasing number of HCMV-BAC derived being generated and used. This is a VERY minor point, which the authors can change or ignore. The statistical analyses throughout the manuscript are appropriate, the manuscript is clearly written, and the referencing is also appropriate. The conclusions that the authors assert are supported by the data, and the newly added data certainly aid in this. The clarity added to the methods is sufficient for additional investigators to reproduce the protocols. Together, this resubmission is greatly improved from the originally submitted manuscript.

Reviewer #2 (Remarks to the Author):

The revised version of this manuscript contains numerous improvements, which increase its quality. A few modifications are however still required to support its publication:

Figure 1. Were the Hoechst and GFP images taken from the exact same fields (an overlay should be provided to show this)? It would not seem so, as nuclei do not seem to correspond to the GFP+ cells. Also, the Hoechst images for Titan-deltaUS28 mock and +F49A-FTP are identical copies of each other. Finally, the data shown in panel B is identical to the one shown in the previous version of the paper, presumably derived from cells infected at an MOI of 1 (?), and not 0.1. Consequently, the sentence "a graphical representation of this data" in the figure legend is wrong. In short: this figure should contain both images AND quantification of cultures infected at an MOI of 1, or images AND quantification of cultures infected at an MOI of 0.1, not a mix and match of figures, quantifications and MOIs.

Figure 4. Panel A: Why is the proportion of latent monocytes surviving F49A-FTP treatment 60% now, when in the previous version of the paper it was 40%? Nothing has changed in the figure legend, so supposedly this data is derived from the same five experiments (?).

Figure S1. Panel A-D: What MOI was/were used to infect cells in each panel (A, B, C and D)? At what times post-infection were the PCR or immunofluorescence assays conducted in each panel (A, B, C and D)?

Figure S3. Panel A: what MOI was used for these infections? Panel B: the actual number of reactivation foci observed at each MOI with and without F49A-FTP treatment should be shown, in addition to the relative change in reactivation events. The number of infected cells present in cultures infected at an MOI of 5 is dramatically different than at an MOI of 0.05, and reactivation events may simply become undetectable. Therefore, the actual number of events observed becomes extremely important and should be reported.

Figure 5: what MOI was used in these experiments? Please add a reference for the differentiation of CD34+ cells into immature dendritic cells using the listed cytokines (G-CSF, TNF- α , GM-CSF and IL-4 at 1000 U/ml for 5 days): has someone shown that this culture method actually yields dendritic cells?

Instead of reporting the MOI used in a single sentence in the Materials and Methods (e.g. "Monocytes and THP-1 were infected with all HCMV viral isolates at a predicted multiplicity of infection (MOI) of 5 (based on RPE-1 cells) leading to 10% infected cells; figure S3B uses dilution of the viral stock, starting at an MOI of 5") it would be immensely better to state the MOI used, the resulting percentage of infected cells and the times of harvest post-infection actually used to generate the data reported in each figure. These details are extremely important and should be reported in each figure legend.

Reviewer #3 (Remarks to the Author):

Authors have appropriately addressed all the issues raised by me. I have no further comments.

We would like to thank all the reviewers for their extremely helpful comments. Our responses to the each reviewer are shown in italics, below.

Reviewers' comments:

Reviewer #1 (Remarks to the Author):

In this resubmission by B.A. Krishna, et al, entitled “Targeting the latent cytomegalovirus reservoir with a novel fusion toxin protein”, the authors show that F49A-FTP specifically targets latently infected hematopoietic cells (CD14+, CD34+, and THP-1) via interaction with US28. This thereby reduces the amount of cells capable of reactivation in response to the appropriate stimuli. As the authors suggest, this could prove a novel mechanism by which to rid (or drastically reduce) a bone marrow graft of HCMV latently infected cells prior to hematopoietic transplantation, thus reducing the possibility for cytomegalovirus reactivation and increasing quality of life for the transplant recipient. The authors have made significant changes to the manuscript from the first review. Importantly, they have alleviated many of the concerns raised by the three reviewers by including some additional experiments as well as clarifying sections of the text. The experiments that the authors added (e.g.: time course RTqPCR data for CX3CR1, Δ US28-infected monocytes +/- F49A-FTP, US28 RTqPCR time course data, added time course analyses of F49A-FTP, effectiveness of F49A-FTP over various MOI's) have significantly improved the manuscript from the first submission. The clarity added to the methods, results, and discussion sections have also greatly improved the manuscript on a whole. To the point of Reviewer #2 regarding the TB40/E variant used herein, it seems appropriate to use the abbreviation that was published in the original manuscript to avoid confusion, as there are an increasing number of HCMV-BAC derived

being generated and used. This is a VERY minor point, which the authors can change or ignore. The statistical analyses throughout the manuscript are appropriate, the manuscript is clearly written, and the referencing is also appropriate. The conclusions that the authors assert are supported by the data, and the newly added data certainly aid in this. The clarity added to the methods is sufficient for additional investigators to reproduce the protocols. Together, this resubmission is greatly improved from the originally submitted manuscript.

We would like to thank the reviewer for their previous helpful suggestions and are pleased that our revisions are acceptable.

Reviewer #2 (Remarks to the Author):

The revised version of this manuscript contains numerous improvements, which increase its quality. A few modifications are however still required to support its publication:

Figure 1. Were the Hoechst and GFP images taken from the exact same fields (an overlay should be provided to show this)? It would not seem so, as nuclei do not seem to correspond to the GFP+ cells. Also, the Hoechst images for Titan-deltaUS28 mock and +F49A-FTP are identical copies of each other. Finally, the data shown in panel B is identical to the one shown in the previous version of the paper, presumably derived from cells infected at an MOI of 1 (?), and not 0.1. Consequently, the sentence “a graphical representation of this data” in the figure legend is wrong. In short: this figure should contain both images AND quantification of cultures infected at an MOI of 1, or images AND quantification of cultures infected at an MOI

of 0.1, not a mix and match of figures, quantifications and MOIs.

We would like to thank the reviewer for noticing this error and apologise for our mistake which resulted from a mix up of panels. We have now included the correct Hoechst stained image for the Titan-deltaUS28 +F49A-FTP panel and, as requested, have merged the images in Figure 1. The merged panels now clearly show the superimposition of GFP+ cells and their nuclei.

We apologise that figure 1B was not updated, this was our mistake. This has been rectified, as requested.

Figure 4. Panel A: Why is the proportion of latent monocytes surviving F49A-FTP treatment 60% now, when in the previous version of the paper it was 40%? Nothing has changed in the figure legend, so supposedly this data is derived from the same five experiments (?).

We would like to thank reviewer for noticing that we had mistakenly used a graph of data after 48 hours of treatment with F49A-FTP - previously, the graphs were from latent monocytes treated with F49A-FTP for 72 hours. We have now used data from the correct time-point and have corrected the statistical analysis for this figure to reflect the correct data set.

Figure S1. Panel A-D: What MOI was/were used to infect cells in each panel (A, B, C and D)? At what times post-infection were the PCR or immunofluorescence assays conducted in each panel (A, B, C and D)?

We have added MOI data into the figure legend, as requested. In order to keep these analyses as comparable as possible, all analyses were performed 4 days post

infection. The exception to this was the staining and RNA analysis for IE and UL99 in dendritic cells, where the monocytes or CD34 cells were infected, given three days to establish latency, and then differentiated for 7 days. Cells were fixed, or RNA was harvested, 4 days post differentiation. We have updated the figure legend to explain the exact time course for each experiment.

Figure S3. Panel A: what MOI was used for these infections? Panel B: the actual number of reactivation foci observed at each MOI with and without F49A-FTP treatment should be shown, in addition to the relative change in reactivation events. The number of infected cells present in cultures infected at an MOI of 5 is dramatically different than at an MOI of 0.05, and reactivation events may simply become undetectable. Therefore, the actual number of events observed becomes extremely important and should be reported.

We have updated the figure legend for Figure S3A to state that these monocytes were infected at an MOI of 5, as requested.

For Figure S3B, we should have pointed out that, as we decreased MOIs of infection, we increased the total number of cells for each analysis to ensure that reactivation events were always detectable. As such, we have updated the methods section to explain this in full.

In addition, in our analysis of killing of naturally latent monocytes by F49A-FTP (Figure 6), we also increased the number of naturally latent monocytes used in the analysis which is now detailed in the figure legend.

Figure 5: what MOI was used in these experiments? Please add a reference for the differentiation of CD34+ cells into immature dendritic cells using the listed

cytokines (G-CSF, TNF- α , GM-CSF and IL-4 at 1000 U/ml for 5 days): has someone shown that this culture method actually yields dendritic cells?

Details about the MOI have been now been included in the figure legend, as requested and we have added a reference (Reeves et. al, JGV 2005), which confirms that this culture method does, indeed, yield dendritic cells and reactivate latent HCMV.

Instead of reporting the MOI used in a single sentence in the Materials and Methods (e.g. “Monocytes and THP-1 were infected with all HCMV viral isolates at a predicted multiplicity of infection (MOI) of 5 (based on RPE-1 cells) leading to 10% infected cells; figure S3B uses dilution of the viral stock, starting at an MOI of 5”) it would be immensely better to state the MOI used, the resulting percentage of infected cells and the times of harvest post-infection actually used to generate the data reported in each figure. These details are extremely important and should be reported in each figure legend.

We agree that this would clear up any confusion, and have updated each figure legend to include MOIs and time of harvesting post infection.

Additionally, for the relevant experiments where we quantified differences in GFP+ cells - figures 4, 5, S1, S2 and S3) - we have detailed the percentage of GFP+, latently infected cells, in the relevant figure legends, as requested.

Reviewer #3 (Remarks to the Author):

Authors have appropriately addressed all the issues raised by me. I have no further comments.

We would like to thank reviewer for their previous helpful suggestions and are pleased that are revisions are acceptable.

REVIEWERS' COMMENTS:

Reviewer #2 (Remarks to the Author):

Authors have addressed all of my issues with their manuscript, which I now consider to be acceptable for publication.